# High-intensity irreversible electroporation targeting intracellular structures enhance tumor ablation in lung cancer models

Hong Bae Kim◯¹, Jin Young Youm², Joon-Mo Yang◯²*, Sung Bo Sim◯³*

1 Department of Biosystems & Biomaterials Science and Engineering, Seoul National University, Seoul, Republic of Korea, 2 Department of Biomedical Engineering, Ulsan National Institute of Science and Technology (UNIST), Ulsan, Republic of Korea, 3 Department of Thoracic and Cardiovascular Surgery, The Catholic University of Korea, Buchen, Republic of Korea

* jmyang@unist.ac.kr (J-MY); sungbo@catholic.ac.kr (SBS)

## Abstract

This study aimed to optimize irreversible electroporation (IRE) parameters to enhance intracellular injury, specifically targeting nuclear and mitochondrial structures that are insufficiently affected by conventional protocols. To address limitations of standard 1000~2500 V/cm clinical settings, we experimentally and computationally evaluated both low- and high-electric-field conditions and identified pulse parameters capable of safely achieving electric field strengths exceeding 4,000 V/cm, values that remain below the predicted arcing threshold for our electrode configuration while permitting effective intracellular electroporation. In vitro studies using A549 lung cancer cells demonstrated that high-field IRE markedly intensified oxidative stress, resulting in a 30-fold increase in hydrogen peroxide production and pronounced disruption of mitochondrial membrane potential. Transmission electron microscopy further confirmed severe ultrastructural injury, including plasma membrane rupture, nuclear membrane deformation, and complete loss of mitochondrial cristae, culminating in irreversible cell death. In vivo experiments corroborated these findings: high-field IRE produced extensive and uniform tumor ablation, whereas conventional lower field strengths generated only localized and partial damage. These results indicate that elevating electric-field intensity in IRE protocols can overcome the inherent limitations of traditional approaches by reliably inducing intracellular organelle damage, suppressing cellular repair pathways, and enhancing overall ablation completeness. Further studies are warranted to evaluate long-term safety and therapeutic durability of high-field IRE in vivo.

## Introduction

Cancer remains a leading cause of morbidity and mortality worldwide, with lung cancer being the most prevalent and aggressive type [1]. Although typical therapies,

**Data availability statement:** All relevant data are within the manuscript.

**Funding:** This research was funded by the National Research Foundation of Korea, grant number 2022R1F1A1075102 and 2022R1F1A1072398, and by Tech Incubator Program for Startup (RS-2023-00303400) of Ministry of SMEs and Startups. The funders had no role in study design, data collection and analysis, decision to publish, or preparation of the manuscript.

**Competing interests:** The authors declare no potential conflicts of interest with respect to the research, authorship, and/or publication of this article.

such as surgery, chemotherapy, and radiation, have been effective, their limitations, including substantial side effects and incomplete tumor ablation, necessitate the development of alternative approaches [2]. Irreversible electroporation (IRE) has gained attention as a non-thermal ablation modality capable of treating tumors located near heat-sensitive structures [3,4].

IRE applies high-voltage, microsecond-duration electric pulses that induce extensive nanopores in the plasma membrane, ultimately disrupting cellular homeostasis and activating downstream apoptotic or necrotic pathways [5,6]. Although the membrane, owing to its capacitive properties, is the primary site of initial pore formation, pulsed electric fields at kilovolt-per-centimeter intensities are known to exert bioelectric stresses that measurably affect intracellular organelles such as mitochondria, chromatin, and cytoskeletal components [7]. This suggests that the distinction between membrane-focused and intracellular electroporation is quantitative rather than categorical, as both cellular compartments can be substantially compromised under clinically relevant IRE parameters.

However, conventional IRE protocols typically employ field strengths of 1000–2500 V/cm. At these amplitudes, substantial spatial heterogeneity can arise within tumors due to variations in tissue conductivity, perfusion, fibrosis, and geometric constraints. As a result, the effective electric field inside the tumor can fall well below the externally applied value, leading to sublethal electroporation at the tumor margins [8,9]. Surviving cancer cells may recover metabolic function and proliferative capacity, underscoring the need for more robust parameter settings that ensure irreversible injury throughout the targeted volume.

Increasing the electric field strength represents a direct strategy to overcome these limitations [10]. Electric fields approaching or exceeding 3000 V/cm have been shown to increase the likelihood of irreversible injury in both membrane and intracellular compartments [11]. Higher fields can generate larger and more persistent pores, accelerate transmembrane potential rise, amplify mitochondrial dysfunction, and promote oxidative stress—key determinants of irreversible cell death [12,13]. Moreover, high-field microsecond pulses may better compensate for intratumoral field attenuation caused by conductivity gradients and tissue heterogeneity, thereby expanding and stabilizing the ablation zone [10].

At the same time, the therapeutic window for high-field IRE is not fully established. Elevated voltages raise concerns regarding sparking, localized heating, and unintended collateral injury, necessitating careful optimization of pulse width, pulse number, electrode spacing, and waveform design [14]. Importantly, microsecond-scale pulses remain within the β-dispersion regime where membrane capacitance shapes intra- and extracellular field distributions, meaning that both membrane and intracellular responses must be considered when increasing field strength [15,16].

High-frequency irreversible electroporation (H-FIRE), which uses short bipolar pulses to reduce muscle contractions, offers advantages for specific procedural contexts [17]. Although clinical use in human tumors remains limited, several pre-clinical in vivo studies have demonstrated effective tumor ablation using H-FIRE waveforms at electric field strengths exceeding 2300 V/cm [18]. However, typical

H-FIRE clinical protocols operate around ~2300 V/cm and do not directly address the need for higher electric fields to enhance intracellular injury or compensate for intratumoral field drop. Therefore, the rationale for exploring high-fields over 3000 V/cm is rooted in the biophysical and tissue-level limitations of conventional IRE rather than waveform-specific innovations.

[18–30] This study aimed to investigate the optimization of IRE parameters, including high electric field intensity, pulse duration, and pulse number, for achieving uniform cell death in lung cancer models. Using *in vitro* and *in vivo* systems, we compared high electric field and typical low electric field one to evaluate their differences in cellular viability, structural integrity, and the extent of intracellular injury. Additionally, we analyzed the role of oxidative stress and mitochondrial dysfunction in driving IRE-mediated cell death.

## Materials and methods

### Numerical simulation on electric field distribution

To precisely assess the electric field distribution, particularly field heterogeneity near electrode surfaces and edges, we used COMSOL Multiphysics (version 5.4; COMSOL, Stockholm, Sweden) to numerically simulate electric field intensity across voltages ranging from 400 to 2,400 V around aluminum plate electrodes (Cuvette Plus, 4-mm gap, 800 lug, BTX, Massachusetts, USA). Although ideal planar electrode configurations theoretically predict the electric field by dividing the voltage by the electrode gap distance, practical experimental conditions introduce complexities such as non-uniformities near electrode edges, partial medial filling, and electrode geometry that markedly distort and intensify local electric fields. Therefore, these simulations aimed explicitly to quantify actual spatial distributions, intensity gradients, and heterogeneities within the cuvette, particularly at electrode edges, where localized high fields concentrations can theoretically approach arcing thresholds. To ensure stable operation at higher voltages in our specific experimental setup, the cuvette was intentionally modeled as half-filled with conductive culture media having an electrical conductivity of 1.7 S/m, thermal conductivity of 0.6 W/Mak, and heat capacity of 4.186 J/Gok at 25 °C based on published values for RPMI-based culture media containing serum [19,31]. This partial fill configuration was intended to maintain electrical stability at higher voltages, reducing the likelihood of approaching arcing thresholds (Fig 1a). In the simulation, the electric field distribution was determined using the Laplace equation under an electrostatic approximation as follows [20]:

$$\nabla \cdot (\sigma \nabla \varphi) = 0 \tag{1}$$

where $\varphi$ denotes the electric potential, and $\sigma$ is the electrical conductivity of the media. This equation essentially states that in a steady-state condition with conductive media, the divergence of the current density ($J = \sigma E = -\sigma \nabla \varphi$) is zero. This is because no net accumulation of charge occurs within the media in a stable system.

For simplicity, our simulation assumed that the electrical properties of the media, including those with cells, remained constant during the 20-pulse application. The mesh for the simulation comprised 221,402 elements. An electric boundary condition was applied on the exposed solid electric surfaces: $\varphi = V$ (source) and $\varphi = 0$ (sink), while other surfaces were defined as electric insulation. The resulting electric field distribution is shown on Fig 1b.

### Cell culture

A human lung cancer cell line (A549) with features of carcinoma histopathology was purchased from the Korean Cell Line Bank (Jongno-guy, Seoul, Korea). The cells were maintained in RPMI-1640 media (Celgene, Gyeongsan-si, Korea) supplemented with 10% fetal bovine serum and 1% antibiotics (penicillin (100 U/mL)-streptomycin (100 go/mL)) purchased from Gibco, Life Technologies and incubated at 37 °C in a humidified environment of 5% $CO_2$ using an incubator (371GP, Thermon Fisher Scientific, Massachusetts, USA).

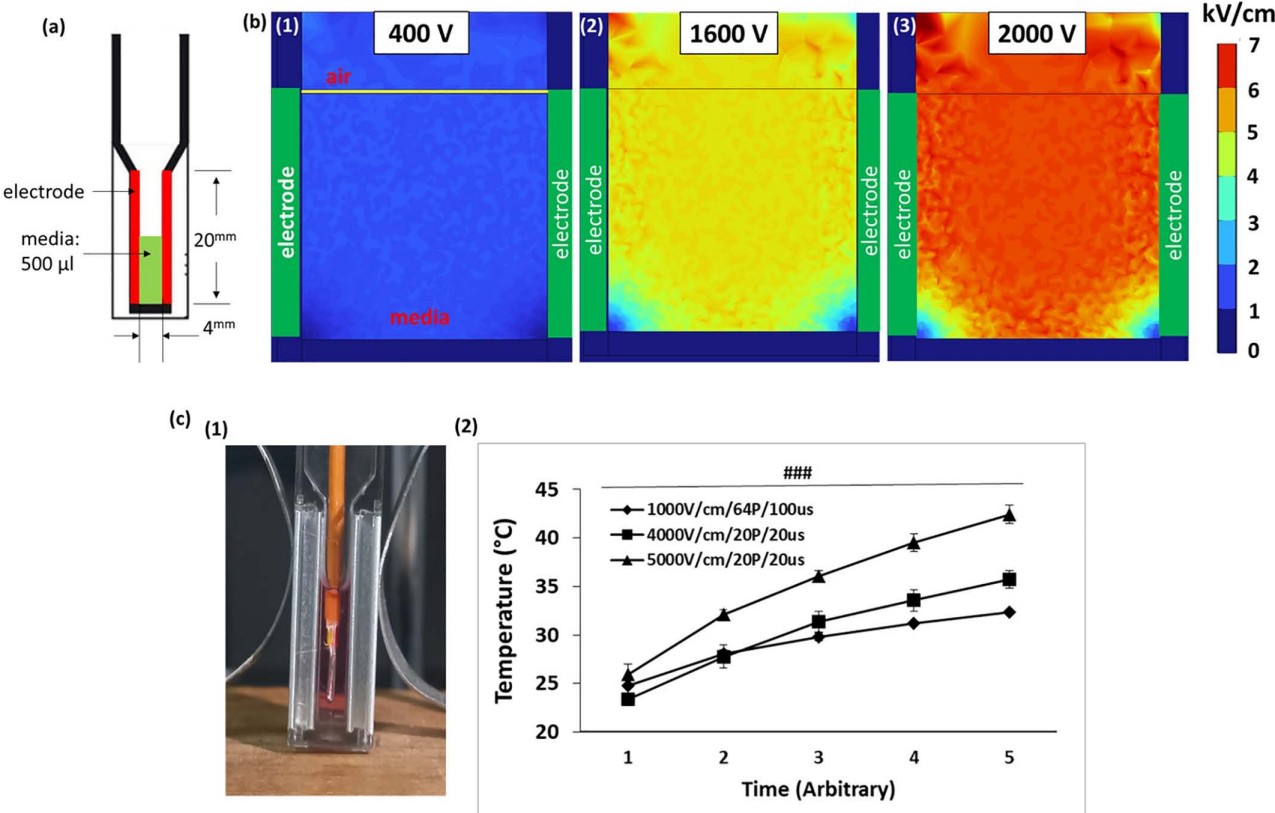

**Fig 1. Simulation of electric field distribution and temperature measurements in culture media. (a)** Schematic of the experimental setup showing a cuvette partially filled with a 500 µL media and two aluminum plate electrodes separated by a 4-mm gap. **(b)** Numerical simulations of electric field distributions at applied voltages of 400 V (1), 1,600 V (2), and 2,000 V (3). Color bar indicates electric field intensity distribution in kV/cm. **(c)** Temperature measurement during IRE procedures. (1) Experimental setup showing the position of a cc-type thermocouple within the media. (2) Measured temperature profiles under different IRE conditions.

## IRE procedure in vitro

Based on these simulations, the external voltages of 400, 800, 1,200, 1,600, 2,000, and 2400 V were applied in the form of rectangular direct current with pulse durations of 10, 20, and 40 is and pulse numbers of 10, 20, and 40 at a frequency of 1 Hz in the culture media in the cuvette. Compared with that of a typical IRE setting, an additional condition using 400 V with a pulse duration of 100 is and 64 pulses was tested and matched to deliver similar electrical energy as that of a 1600V/20P/20µs condition. For outputting such powers, a pulse generator (BTX940, BTX, Massachusetts, USA) that produces a square wave voltage of up to 3,000 V was used. The generator was linked with the cuvette to deliver electrical energy through a jig (Safety dome for X2, BTX) for the cuvette. To exclude experimental conditions that might cause sparking, the above conditions were decreased in ≤ 800-µL culture media. The spark was not observed in the media of 500 µL under 2,000 V except for 2,400 V. Subsequently, the harvested cells of $5.0 \times 10^4$ in the culture media of 500 µL were transferred into the cuvette, and IRE was applied according to the conditions. Electrical resistance was assessed using the BTX generator's built-in Pretest function, which applies a fixed, low-voltage test pulse before and after electroporation to determine baseline and post-pulse resistance under voltage-independent conditions. Because resistance measured directly after IRE pulses is strongly dependent on the applied voltage amplitude, it cannot be used for cross-condition comparison. In contrast, the Pretest pulse uses the same voltage for all conditions, allowing resistance

values to be compared consistently regardless of the electroporation setting. This approach ensures that observed resistance differences reflect changes in the sample rather than differences in applied voltage. Cells remained in the cuvette approximately 8–10 sec longer than the pulsing period, accounting for the parameter-setting period and additional handling. All parameters were summarized in Table 1.

To assess potential thermal effects during pulsing, temperature changes in the culture media were measured directly during pulse application under representative experimental conditions. Temperature measurements were conducted using a digital thermometer (Fluke 53IIB, Fluke Corporation, WA, USA) with a cc-type thermocouple sensor. The sensor was carefully positioned near the midpoint of the media within the cuvette, as depicted in Fig 1c, ensuring the thermocouple tip was fully immersed in the media without contacting the aluminum electrodes. Continuous temperature readings were recorded at intervals corresponding to the pulse delivery sequence, allowing quantification and comparison of temperature increments associated with each pulse protocol (Fig 1c).

## Cell viability

Cell viability after IRE treatment was quantitatively assessed using the Cell Counting Kit-8 (CCK-8; Darmstadt, Germany). Briefly, immediately following IRE treatment, cells were transferred into a 24-well plate at a density of $5.0 \times 10^4$ per well containing 500 µL of culture media and incubated for 24 h at 37 °C. Next, the CCK-8 reagent, containing a tetrazolium salt WST-8, which is enzymatically reduced by metabolically active cells to generate a soluble, orange-colored formazan dye, was added directly to each well at a volume ratio of 1:10 (50 µL per well). After incubating at 37 °C for 2 hours, the absorbance was measured at 450 nm using a microplate reader (Versa ax, Molecular Devices, CA, USA). The absorbance values were proportional to the number of metabolically active, viable cells.

## Measurement of Annexin V and propidium iodide staining

Cells were harvested and resuspended at a density of $5.0 \times 10^4$ cells in culture media immediately after IRE treatment and incubated for 24 h at 37 °C. At 24 h post-treatment, apoptosis and necrosis were quantified using an Annexin V & Dead Cell assay kit (MCH100105, Luminex, Texas, USA) according to the manufacturer's instructions. Briefly, the cell suspension of 100 µL was transferred into a 1.5-mL microcentrifuge tube and combined with 100 µL of Annexin V & Dead

**Table 1. Summary of electric pulse parameters and energy indices used in high- and low-electric-field conditions at 1 Hz.**

| Voltage (V/cm) | | Pulse width (µs) | Energy index $V^2 \cdot t \cdot N$ ($V^2 \cdot sec$) | | |
|---|---|---|---|---|---|
| | | | Pulse number | | |
| | | | 10 | 20 | 40 |
| HEF | 1000 | 10 | 100 | 200 | 400 |
| | 2000 | 10 | 400 | 800 | 1600 |
| | 3000 | 10 | 900 | 1800 | 3600 |
| | 4000 | 10 | 1600 | 3200 | 6400 |
| | | 20 | 0 | 6400 | 0 |
| | | 40 | 0 | 12800 | 0 |
| | 5000 | 10 | 2500 | 5000 | 10000 |
| | 6000 | 10 | 3600 | 0 | 0 |
| LEF | | | Pulse number | | |
| | | | 64 | | |
| | 1000 | 100 | 6400 | | |

Applied all to be frequency 1 Hz

Cell reagent. The mixture was gently pipetted to ensure homogeneity and incubated at 37 °C for 20 min. Cells were then analyzed using a Muse Cell Analyzer (Cytec Guava Muse, EMD Millipore, Massachusetts, USA). Cells positive for both Annexin V and propidium iodide (PI) were classified as late apoptotic, and cells negative for Annexin V but positive for PI were classified as necrotic. The apoptotic rate reported in this study refers to the combined percentage of early and late apoptotic cells.

## Measurement of intracellular nitrogen oxide and hydrogen peroxide levels

Intracellular nitrogen oxide (NO) levels were measured using the NO Assay Kit (ab272517, Abcam, Cambridge, UK). Cells in 500 μL of the culture media were exposed to experimental conditions and immediately transferred into a 24-well plate where they were suspended at $5.0 \times 10^4$ cell/well and incubated for 24 h. Following harvest, centrifugation, and resuspension in phosphate-buffered saline, the cells were deproteinized using $ZnSO_4$ and NaOH. Subsequently, samples and serially diluted nitrite standards were mixed with the working reagent (Reagents A, B, and C) and incubated for 10 min at 60 °C. After incubation, reaction tubes were briefly centrifuged, and 250 μL of the supernatant was transferred into a 96-well plate for optical density (OD) measurement at 540 nm using a Versa ax microplate reader (Molecular Devices, Silicon Valley, CA, USA). Results were quantified by comparing the OD values to a standard curve generated from known NO concentrations, enabling the precise measurement of NO levels as a marker of oxidative stress following various IRE conditions.

Furthermore, cells were subjected to experimental conditions to induce oxidative stress and seeded at $5 \times 10^4$ cells in 500 μL of the culture media in a 24-well plate, followed by a 24-h incubation for attachment and recovery. The culture media were removed from all wells, and ROS-Glo™ $H_2O_2$ substrate (Promega G8820) was added, with the plate incubated for 2 h at 37 °C in a $CO_2$ incubator to allow substrate interaction with cellular $H_2O_2$. Subsequently, a reactive oxygen species-Glo Detection Solution of 100 μL was added to each well, and the plate was incubated at room temperature for 20 min. The media samples of 150 μL were transferred into a separate opaque white plate to prevent light interference during measurement, and relative luminescence units were recorded using a Glom ax Navigator GM2000 microplate reader, with each well-read for 0.3 s. This protocol facilitated the detection of $H_2O_2$ as an oxidative stress marker, quantifying cellular responses to various IRE conditions.

## Measurement of the mitochondrial membrane potential

Cells were prepared for mitochondrial membrane potential (MMP) assessment by suspending them at $5.0 \times 10^4$ cells per 500 μL culture media within the cuvette. After exposure to the designated experimental conditions, the cells were immediately harvested and processed using the Muse® Mito Potential kit (Luminex) to assess mitochondrial depolarization. Briefly, the dye provided in the kit was diluted to a 1:1,000 ratio in a 1× assay buffer to create a working solution. For each sample, 95 μL of the diluted Mito Potential dye and 5 μL of Muse Mito Potential 7-AAD reagent were combined with the cells. Samples were incubated for 20 min at 37 °C, followed by an additional 5 min incubation at room temperature. Subsequently, the Muse Cell Analyzer (Cytec Guava Muse) quantified mitochondrial membrane depolarization by analyzing fluorescence intensity shifts. Data analysis involved distinguishing cell populations with intact or depolarized mitochondrial membranes based on their fluorescence profiles, thus enabling precise measurement of mitochondrial depolarization induced by IRE conditions.

## Measurement of the cytoplasmic membrane potential

The fluorescent dye FLIPR-RED (FLIPR, Molecular Device, California, USA) measured the cytoplasmic membrane potential (CMP). Cells of $2.0 \times 10^5$ were first applied according to the experimental conditions, resuspended, and allowed to recover for 30 min to ensure complete resealing of membrane pores in cuvettes. Following the recovery period, cells were

incubated with 1% FLIPR-RED dye in the culture media for 30 min. Subsequently, the CMP was assessed through flow cytometry using the Muse open module with a yellow filter (emission wavelength of 576 nm). The results were analyzed and compared with those of a control group.

## Measurement of intracellular calcium ions

Intracellular calcium concentration was measured using the fluorescent calcium indicator Fluo-4 AM (Molecular Probes, Invitrogen, USA). After electroporation, $2.0 \times 10^5$ cells were resuspended in cuvettes. Subsequently, the cells were incubated with Fluo-4 AM (5 μM) in the culture media for 35 min at 37 °C. Following incubation, the cells were washed thrice with a warm Hank's balanced salt solution to remove extracellular dye. Next, the cells were resuspended in a fresh HBSS and reintubated for 30 min at 37 °C to allow the complete de-esterification of intracellular Fluo-4 AM. The calcium levels were assessed through flow cytometry, with the excitation and emission spectra of the Fluo-4 being measured at 494 and 576 nm, respectively, using a laser.

## Measurement of DNA damage

The DNA damage was analyzed by determining phosphorylated ataxia-telangiectasia mutation (ATM) and H2A histone family member x (H2A.X) (MCH200107 multi-color DNA damage kit, Luminex, USA), which are indicators of DNA double-strand breaks and activation of the cellular DNA damage response pathway. Cells were prepared at $2 \times 10^5$ cells 500 μL of culture media in the cuvette and treated with IRE conditions. Following 24-h incubation, cells were centrifuged and washed once with 1 × PBS and fixed with a fixation buffer (provided by the kit) on ice for 10 min. Cells were subsequently permeabilized with a permeabilization buffer on ice for an additional 10 min, washed again with 1 × PBS, and resuspended in a 1 × assay buffer. For antibody staining, 10 μL of a 20 × antibody cocktail containing ATM and H2A.X was added to 90 μL of a 1 × assay buffer with the cells, and the samples were incubated in the dark at room temperature for 30 min. After staining, cells were washed and resuspended in 200 μL of a 1 × assay buffer for analysis. The percentage of cells positive for phosphorylated ATM and H2A.X were quantified separately using the Muse Cell Analyzer (Cytec Guava Muse). The total DNA damage presented in this study refers to the combined percentage of cells positive for phosphorylated ATM and/or H2A.X, providing an integrated measure of the DNA damage response induced by IRE treatment.

## Expression of polymerase chain reaction

Following IRE, $5.0 \times 10^5$ cells were collected after a 24-h pulsing. The cells were washed twice with PBS and lysed using the RNeasy Plus Mini Kit (Qiagen, Hilden, Germany) to extract total RNA. Equal amounts of RNA from each sample were converted to cDNA using the 5 × cDNA Synthesis Master Mix (Cell safe, Gyeonggi-do, Korea) with reverse transcriptase and random primers, according to the manufacturer's protocol. The resulting cDNA was subjected to real-time PCR on a CFX96™ Real-Time System (Bio-Rad, California, USA) to quantify gene expression. Each qPCR reaction included Light Cycler 480 SYBR Green I Master (Roche, Mannheim, Germany), gene-specific primers, and a cDNA template. The thermal cycling protocol started with an initial denaturation at 95 °C for 10 min, followed by 40 cycles at 95 °C for 15 s, 60 °C for 30 s, and 72 °C for 30 s, with melt curve analysis for specificity. Relative gene expression was calculated using the comparative cycle-threshold (CT) method, normalized to the housekeeping gene CIAO1, and expressed as fold change relative to control. Primer sequences targeting genes involved in apoptosis and necrosis enabled an assessment of gene expression changes in response to IRE treatment (Table 2).

## In vitro transmission electron microscopy

For the transmission electron microscopy (TEM), cells were seeded at $2.0 \times 10^5$ in a 35-mm culture dish and allowed to adhere overnight. Electroporation performed at an applied voltage of 200 and 800 V, generating an electric field strength

**Table 2. Gene sequence for polymerase chain reaction.**

| Gene | Sequence (5′–3′) |
|---|---|
| ATM | F: CTGTGGTGGAGGGAAGATGT, R: GTTGATGAGGGGATTGCTGT |
| ATR | F: GCTCCGATCGTGTACAAATG, R: CACACGCATGGGATAAGAT |
| CHK1 | F: TGTCAGAGTCTCCCAGTGGA, R: AGGGGCTGGTATCCCATAAG |
| CHK2 | F: GGCTTCAGGATGAAGACATGA, R: CACAACACAGCAGCACACAC |
| PARP1 | F: GCTCCTGAACAATGCAGACA, R: TCCTGATGATCTCGGCTTCT |
| BRCA | F: TCCTGATGATCTCGGCTTCT, R: ACTCTGGGGCTCTGTCTTCA |
| P53 | F: TAACAGTTCCTGCATGGGCGGC, R: AGGACAGGCACAAACACGCACC |
| BAX | F: AACATGGAGCTGCAGAGGAT, R: CAGTTGAAGTTGCCGTCAGA |
| CIAO1 | F; TTGGGTCTGGGAAGTTGATGA, R: ACTCTGGGGCTCTGTCTTCA |

of 1,000 and 4,000 V/cm, respectively, using electrodes (Petri pulser (45–0130), BTX, Massachusetts, USA). Immediately following electroporation, cells were fixed within approximately 10 sec after the last pulse to minimize the influence of secondary cellular responses, such as apoptosis or reversible membrane recovery processes. Fixation was achieved by immersing cells in a pre-warmed fixation buffer at 37 °C containing 2% paraformaldehyde and 2.5% glutaraldehyde in 0.15 M sodium cacodylate buffer of pH 7.4. Subsequently, the cells were placed on ice for 1 hour to optimize mitochondrial structural preservation and membrane contrast. Post-fixation was performed using 1% osmium tetroxide solution, 0.8% potassium ferrocyanide, and calcium chloride of 3 mM in a cacodylate buffer of 0.1 M for 1 h. The cells were washed in ice-cold distilled water and stained with 2% uranyl acetate at 4 °C to enhance contrast. After dehydration through a graded ethanol series, the cells were embedded in Durcan resin (Fluka, St. Louis, MO, USA). Ultrathin sections were cut in 70 nm and post-stained with uranyl acetate and lead citrate to improve electron density. Images were acquired using an HT7800 transmission electron microscope (Hitachi, Japan) operating at 80 kV. Micrographs were digitized at 1,800 dpi using a Nikon Cool Scan system, resulting in a resolution of 1.77 nm per pixel. Specific cellular ultrastructural features, including plasma membrane rupture, nuclear envelope deformation, and mitochondrial cristae loss, were evaluated to correlate damage severity with the applied electroporation conditions.

## In vivo TEM

This study was approved by the Institutional Animal Care and Use Committee of Catholic Medical Center (20-004). All methods were performed in strict accordance with the ARRIVE guidelines and the US National Institutes of Health standards for the care and use of laboratory animals, ensuring compliance with all relevant guidelines and regulations. The experiment aimed to analyze cellular ultrastructural changes following IRE using TEM. Three 5-week-old male BALB/c nude mice (Central Lab. Animal Inc., Seoul, Korea) weighing 15–20 g was housed individually under a 12-h light/dark cycle at a controlled temperature of 24 ± 1°C and 55 ± 10% humidity, with free access to food and water. Following a 7-day acclimatization, cultured human cancer cells ($3.0 \times 10^6$ cells in 0.1 mL of Matrigel) were subcutaneously injected into the right flank of each mouse. Once tumors reached a diameter of 7 mm, the mice were anesthetized with an intramuscular injection of 50 mg/kg zolazepam and tiletamine (Soleil 50; Virbac, Carros, France) and 10 mg/kg xylazine (Rompuy; Bayer HealthCare, Leverkusen, Germany). Adequate depth of anesthesia was confirmed by the absence of the pedal withdrawal reflex. To minimize discomfort, meloxicam of 6 mg/kg in s.c. was administered for analgesia, and animals were monitored continuously for respiration, movement, and signs of pain or distress. Electrodes with a 0.8 mm diameter and a 7.5 mm inter-electrode distance were inserted into the tumor to a depth of 10 mm. Electric pulses were applied using two conditions: 3,000 V to achieve an electric field strength of 4,000 V/cm and 750 V to achieve 1,000 V/cm, mirroring the parameters used in the *in vitro* tests. Following pulsing, tissue samples were collected from areas near the anodic electrode to

facilitate the observation of intracellular ultrastructural changes. TEM sample preparation followed the same protocols as those in the *in vitro* experiments, ensuring consistent fixation, post-fixation, staining, and imaging procedures. After completion of the experiment, mice were euthanized using $CO_2$ inhalation followed by cervical dislocation to ensure death. To minize distress, animals were closely monitored, and humane endpoints were implemented as predefined criteria.

## Statistical analysis

All experiments were repeated at least thrice, and data were expressed as the mean ± standard deviation. For comparisons involving more than two groups, statistical significance was assessed using a one-way analysis of variance (ANOVA) followed by Tukey's HSD post-hoc test. For comparisons involving only two groups, an unpaired two-tailed Student's t-test was used. All statistical analyses were performed using Excel software. To clearly distinguish between test types in the figures, ANOVA/Tukey post hoc results are indicated by symbols (#, ##, ###), whereas **t-test results are indicated by asterisk (*, **, ***). Statistical significance thresholds were defined as: #$p < 0.05$, ##$p < 0.01$, ###$p < 0.001$ (ANOVA + Tukey HSD); *$p < 0.05$, **$p < 0.01$, and ***$p < 0.001$(t-test).

## Results

### Simulation of electric field distribution and thermal test for optimal IRE in the culture media

Fig 1 showed that the electric field distribution and thermal happening in the cuvette. The numerical simulations demonstrated the distribution and intensity of the electric field generated at different applied electric field strengths of 1000 V/cm (400 V), 4000 V/cm (1600 V), and 5000 V/cm (2000 V) within the electroporation cuvette setup (a). Electric field intensity notably increased with increasing voltage, especially in regions near the electrode edges. A relatively uniform and low-intensity electric field distribution was observed at the electric field strength of 1000 V/cm, which was insufficient to achieve the critical electric field strength necessary for effective IRE. At higher voltages of 1600 V and 2000 V, a marked intensification of the electric field was observed, with localized values exceeding 4000 V/cm as indicated by the red color zones. The high field intensity at these voltages was predominantly concentrated near electrode surfaces and edges, which could effectively induce cellular ablation. Moreover, based on the simulation result at 6000 V/cm, an experiment was conducted whether the critical sparking occurs or not in the cuvette environment. The spark intermittently happened between electrodes of the cuvette. Thus, all experiments were conducted under 5000 V/cm.

Additionally, experimental validation confirmed that temperature within the media increased proportionally with applied electric field strength (b). Under the typical low-electric-field of 1000 V/cm, the temperature rose from approximately 25 °C to around 32 °C. Conversely, high-electric-field conditions significantly elevated media temperature, with the 4000 V/cm condition increasing the temperature to approximately 36 °C and the 5000 V/cm condition further elevating it to about 42 °C. Here, the x-axis unit was divided into five intervals for both the 20 sec duration with 20 pulses and the 64 sec duration with 64 pulses.

### Optimization of IRE parameters for effective cell ablation in A549 lung cancer cells

The cell viability and apoptosis revealed that the external electric field, pulse duration, and pulse number significantly impacted the effectiveness of IRE in A549 cells (Fig 2). Increasing the electric field strength from 1000 to 5000 V/cm at a fixed duration of 10 μs caused a notable reduction in cell viability, particularly when increasing pulses (1). The viabilities of 4000 and 5000 V/cm were almost similar each other due to reaching saturation effect. We thus selected 4000 V/cm with 20 pulses for this proper experiment. At a fixed 4000 V/cm and 20 pulses, long pulse durations of 10, 20 and 40 μs further decreased viability (2). Viabilities of 20 and 40 pulses appeared similar, indicating a plateau in our specific experimental configuration: however, this does not represent the general saturation behavior reported in previous IRE and H-FIRE studies [21]. We thus selected 20 μs as the optimal pulse duration. Comparisons across different IRE conditions showed that the high electric field setting of 4000V/cm/20P/20μs (HEF) resulted in similar cell viability to that of the low electric field

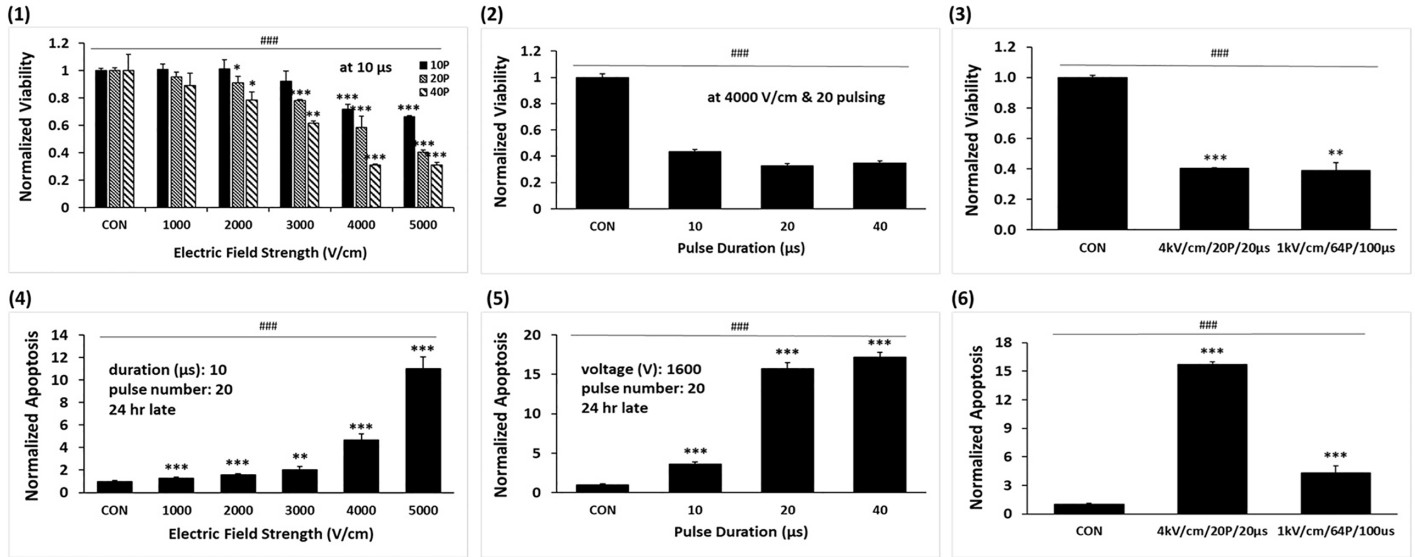

**Fig 2. Cell Viability and apoptosis under various IREs in A549 cells.** (1) Cell viability after various electric field intensity treatment with pulse counts of 10, 20, and 40 pulses fixed at 20μs. (2) Cell viability at a fixed electric field of 4000 V/cm and 20 pulses, with pulse durations of 10, 20, and 40 μs. (3) Comparison of cell viability between 1000 V/cm and 4000 V/cm. (4) Apoptosis across increasing electric field strengths at a fixed pulse duration of 10 μs and pulse number of 20. (5) Apoptosis at electric field strength of 4,000 V/cm and 20 pulses with varying pulse durations of 10, 20, and 40 μs. (6) Comparison of apoptosis between 1000 V/cm and 4000 V/cm. Data are represented as mean ± SEM (n = 3). Multiple-group comparisons were analyzed using one-way ANOVA followed by Tukey's HSD post hoc test, with significance indicated as: $^{###}p < 0.001$. Two-group tests were analyzed using unpaired two-tailed t-tests, with significance indicated as: $^{*}p < 0.05$, $^{**}p < 0.01$, $^{***}p < 0.001$.

strength of 1000V/cm/64P/100μs (LEF) as a typical electric field intensity, even when the total electric energy delivered was the same (3). The apoptosis followed a similar trend, increasing significantly with the electric field strength (4) and pulse duration (5). However, the high-electric field setting induced more apoptosis than the low-electric field one (6).

## Comparative effects of LEF and HEF conditions on membrane integrity and permeability in A549 Cells

Fig 3 compared the effects of HEF and LEF on membrane integrity and permeability in A549 cells, with both conditions delivering the same total energy. Electrical resistance decreased after pulsing across all conditions, and the magnitude of resistance change increased with electric-field strength. Because the cell density was below the threshold required to detect membrane resistance changes, these resistance trends reflect bulk medium conductivity rather than membrane properties. A larger resistance change was observed under HEF than LEF, despite equal total energy delivery. The slight increase in the CMP at 4000 V/cm, however, indicates that HEF may induce strong membrane polarization and destabilization, potentially increasing susceptibility to additional stress (2). Both conditions significantly increased cytosolic calcium levels, possibly because of calcium influx through permeabilized membranes or release from intracellular stores (3). Notably, no significant differences were observed in calcium dysregulation between the two conditions. These findings suggest that while HEF and LEF achieve comparable levels of membrane disruption and permeability enhancement, HEF may offer a slight advantage in membrane destabilization, making it potentially more effective for applications requiring heightened cell membrane sensitivity.

## Comparative effects of HEF and LEF on mitochondrial destabilization and oxidative stress in A549 cells

In Fig 4, changes in MMP (1) and $H_2O_2$ production (2) exhibited minimal changes in lower electric fields. In contrast, high electric field conditions of 4000 and 5000 V/cm caused significant increases in MMP to be 10-fold and 11-fold, respectively

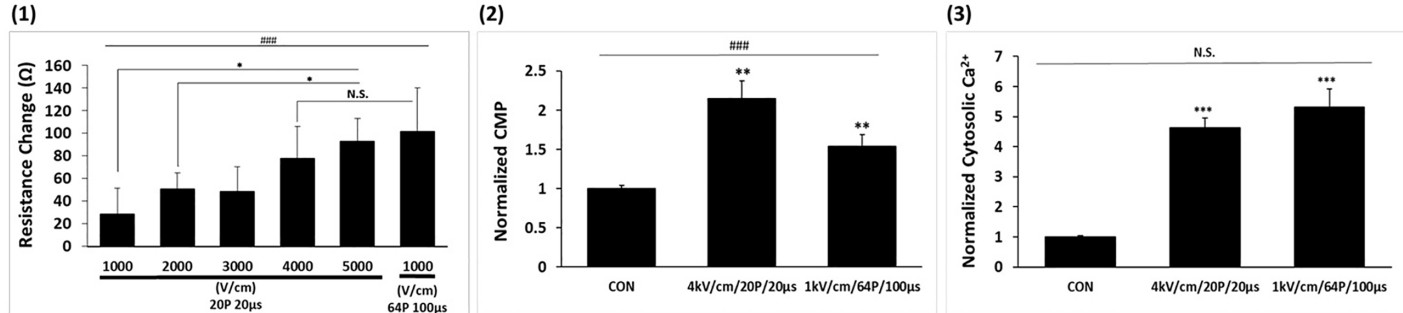

**Fig 3. The electrophysiological responses of A549 cells.** (1) Change in electrical resistance (Ω) in response to different electric field strengths. (2) Cytomembrane potential (CMP). (3) Cytosolic calcium concentration. Data is represented as mean ± SEM (n = 3). Multiple-group comparisons were analyzed using one-way ANOVA followed by Tukey's HSD post hoc test, with statistical significance indicated as: ###$p < 0.001$; N.S, not significant. Two-group tests were analyzed using unpaired two-tailed t-tests, with significance indicated as: *$p < 0.05$, **$p < 0.01$, ***$p < 0.001$.

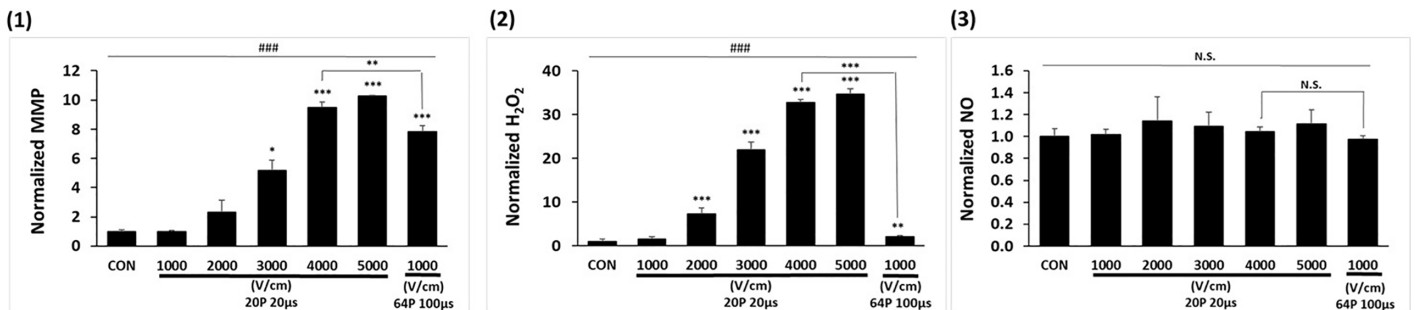

**Fig 4. Impact of variable IRE parameters on mitochondrial dysfunction and oxidative stress.** (1) Mitochondrial membrane potential (MMP). (2) $H_2O_2$ production. (3) NO production. Data is represented as mean ± SEM (n = 3). Multiple-group comparisons were analyzed using one-way ANOVA followed by Tukey's HSD post hoc test, with statistical significance indicated as: ###$p < 0.001$; N.S, not significant. Two-group tests were analyzed using unpaired two-tailed t-tests, with significance indicated as: *$p < 0.05$, **$p < 0.01$, ***$p < 0.001$.

(1) and $H_2O_2$ production to be 30-fold and 35-fold, respectively **(2)**. The LEF achieved MMP changes comparable to those of the HEF (1), albeit with reduced $H_2O_2$ production (2). NO production remained unaffected across all conditions (3).

## Comparative analysis of LEF and HEF inducing DNA damage

Although both conditions, LEF and HEF, delivered the same total electric energy, their effects on DNA damage differed significantly owing to variations in electric field intensity and pulse duration (Fig 5). HEF caused approximately a 3-fold increase in DNA damage compared with that of the control, while LEF achieved a 2.5-fold increase.

## Comparative analysis of LEF and HEF in differential gene expression responses

Fig 6 revealed distinct transcriptional responses under varying IRE conditions, highlighting the crucial processes involved in DNA damage sensing, repair, and apoptosis. Ataxia telangiectasia mutated (ATM) (1), a sensor of double-strand DNA breaks, displayed significant upregulation under 3000 V/cm. In contrast, ATM expression was minimal under 4000 and 5000 V/cm, possibly owing to cellular injury that impaired transcriptional activity. Ataxia telangiectasia and rad3-related protein (ATR) (2), a marker for single-strand DNA damage repair, showed robust upregulation under 3000 V/cm and LEF,

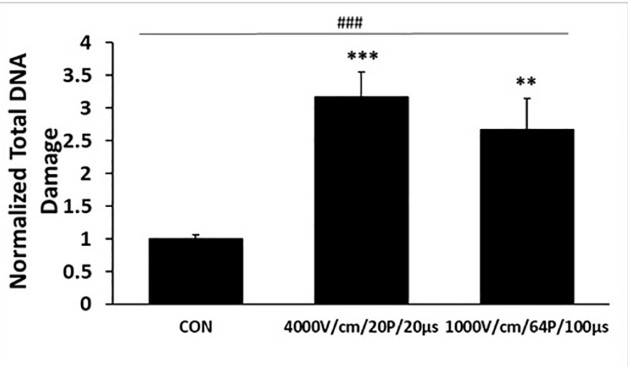

**Fig 5. Evaluation of total DNA damage in A549 cells.** Total DNA damage was quantified by the combined percentage of cells positive for phosphorylated ATM, and H2A.X. Data represent mean ± SEM (n = 3). Multiple-group comparisons were analyzed using one-way ANOVA followed by Tukey's HSD post hoc test, with statistical significance indicated as: $^{\#\#\#}p < 0.001$. Two-group tests were analyzed using unpaired two-tailed t-tests with significance indicated as: $^{**}p < 0.01$, $^{***}p < 0.001$.

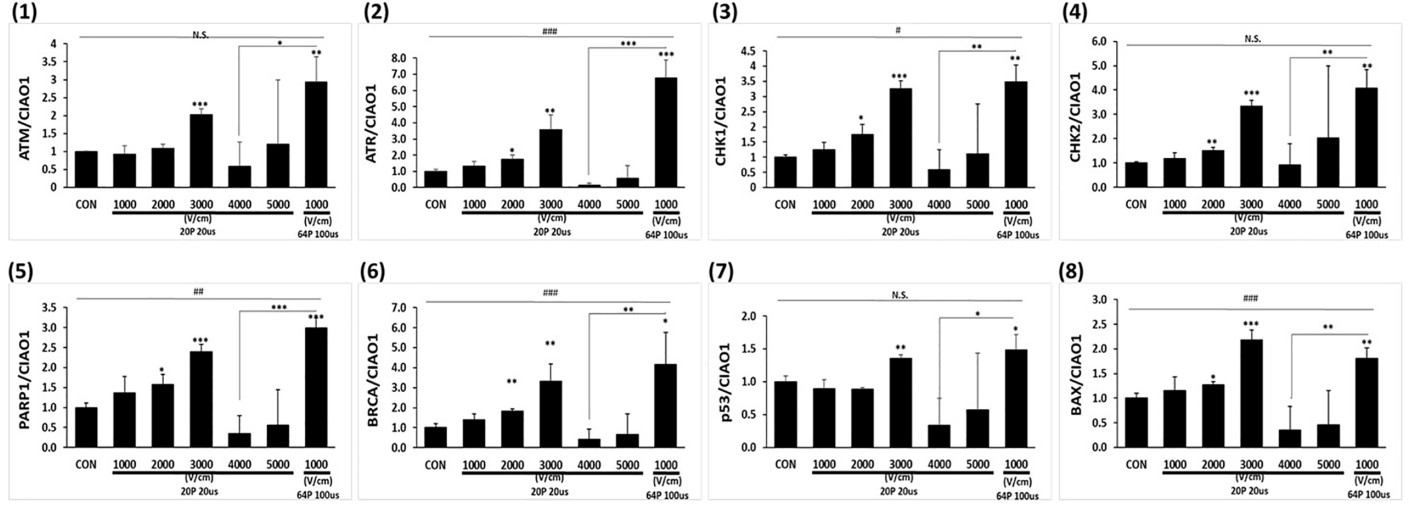

**Fig 6. Differential gene expression analysis in A549 cells.** Real-time PCR showing the relative mRNA expression levels normalized to the reference gene CIAO1, involved in DNA damage sensing (ATM (1) and ATR (2)), stress response coordination (CHK1 (3), CHK2 (4)), DNA repair (PARP1 (5), and BRCA1 (6)), and apoptosis (p53 (7) and BAX (8)). Data are presented as mean ± SEM (n = 3). Multiple-group comparisons were analyzed using one-way ANOVA followed by Tukey's HSD post hoc test, with statistical significance indicated as: $^{\#}p < 0.05$, $^{\#\#}p < 0.01$, $^{\#\#\#}p < 0.001$; N.S, not significant. Two-group tests were analyzed using unpaired two-tailed t-tests, with significance indicated as: $^{*}p < 0.05$, $^{**}p < 0.01$, $^{***}p < 0.001$.

with the latter eliciting a stronger response. In contrast, HEF resulted in negligible ATR expressions, reflecting the suppression of transcriptional activity in cells exposed to extreme damage.

Stress signaling markers checkpoint kinase 1 (CHK1) (3) and checkpoint kinase 2 (CHK2) (4), downstream effectors of ATR and ATM, respectively, exhibited differential responses across the conditions. CHK1 (3) showed peak expression under 3000 V/cm, highlighting an effective activation of ATR-mediated pathways that coordinate single-strand repair and cell cycle arrest. Under LEF, CHK1 exhibited 3.5-fold upregulation compared to control, consistent with the activation of single-strand damage responses in cells exposed to extended pulses. CHK2 (4), responsible for ATM-mediated stress signaling, showed 3.3-fold upregulation under 3000 V/cm and substantial expression under LEF, indicating that both

conditions activate ATM-dependent double-strand break repair. However, under high electric field of 4000 and 5000 V/cm, CHK1 and CHK2 exhibited negligible expression, suggesting a lack of transcriptional activity owing to irreversible damage.

DNA repair markers poly polymerase 1 (PARP1) (5) and breast cancer type 1 susceptibility protein (BRCA1) (6) showed significant expression differences across the conditions. PARP1 (5), which is crucial for base excision repair, exhibited 2.4-fold upregulation under 3000 V/cm, with 3.0-fold expression observed under LEF. BRCA1 (6), essential for homologous recombination repair, followed a similar trend, with 3.3-fold upregulation under 3000 V/cm and peak expression under LEF, highlighting the significance of pulse duration in activating repair mechanisms. Under high electric field of 4000 and 5000 V/cm, PARP1 (5) and BRCA1 (6) showed minimal expression, reflecting the inability of severely damaged cells to engage repair pathways.

Apoptotic markers tumor protein p53 (7) and bcl-2 associated x protein (BAX) (8), critical regulators of stress response and programmed cell death, also showed distinct expression patterns. p53 (7) was significantly upregulated under 3000 V/cm, indicating strong activation of the DNA damage response. BAX (8), a pro-apoptotic gene downstream of p53, exhibited a peak expression under 3000 V/cm, reflecting efficient initiation of apoptosis. Under LEF, p53 (7) exhibited 1.5-fold upregulation compared to control, while BAX (8) showed 1.8-fold expression, demonstrating that LEF effectively activated apoptotic pathways. In contrast, p53 (7) and BAX (8) exhibited minimal expression under 4000 and 5000 V/cm, suggesting that excessive damage bypassed the transcriptional activation of apoptotic pathways and possibly led to necrosis.

### Ultrastructural damage in tumor cells following in vitro under LEF and HEF

Fig 7 illustrated the differences in ultrastructural damage to the plasma membrane, nuclear membrane, and mitochondria under different IRE conditions, highlighting the mechanisms of cellular injury. Compared to control (1 & 2), under HEF (3 & 4), the plasma membrane showed severe rupture and discontinuity, indicating a marked degree of permeabilization and structural instability caused by the intense electric field. The nuclear membrane exhibited pronounced inflation and irregularities, reflecting stress and potential functional damage to the nucleus. The mitochondria were severely damaged, with swollen structures and complete crista loss, suggesting mitochondrial dysfunction and the initiation of apoptosis or necrosis. Under LEF, localized plasma membrane rupture was not observed, and the overall structural integrity was relatively preserved. The nuclear membrane showed mild inflation but remained largely intact with minimal deformation. The mitochondria exhibited partial crista disruption; however, their overall structure was retained, suggesting partial mitochondrial functionality. These results demonstrate that HEF overwhelm cellular repair mechanisms, inducing rapid and irreversible cell death, making them suitable for complete cell ablation, such as in cancer treatment.

### Ultrastructural changes in tumor cells following in vivo under LEF and HEF

To corroborate the ultrastructural changes in an *in vivo* test, TEM was performed for tumor tissues (Fig 8). Compared to control (1 & 2), under HEF (3 & 4), extensive cellular damage was evident. The plasma membrane exhibited severe rupture and discontinuity, indicating high levels of permeabilization and loss of structural stability owing to the intense electric field. The nuclear membrane showed pronounced inflation and irregularities, reflecting nuclear stress and potential functional impairment. The mitochondria displayed pronounced swelling and complete crista loss, indicative of mitochondrial dysfunction and the activation of cell death pathways such as apoptosis or necrosis. In contrast, cells subject to LEF (5 & 6) exhibited comparatively less severe damage. While localized plasma membrane disruptions were observed, the overall integrity of the membrane was better preserved than that under HEF. The nuclear membrane showed mild deformation and inflation but remained largely intact, with less evidence of stress. Mitochondria exhibited partial crista loss but retained their overall structure, suggesting partial mitochondrial functionality and a reduced degree of cellular injury compared with those under high electric field conditions. These findings demonstrate an electric field-dependent response in tumor cells to IRE, where HEF induces extensive and irreversible damage suitable for complete tumor ablation, while LEF results in controlled damage that may preserve some cellular functions.

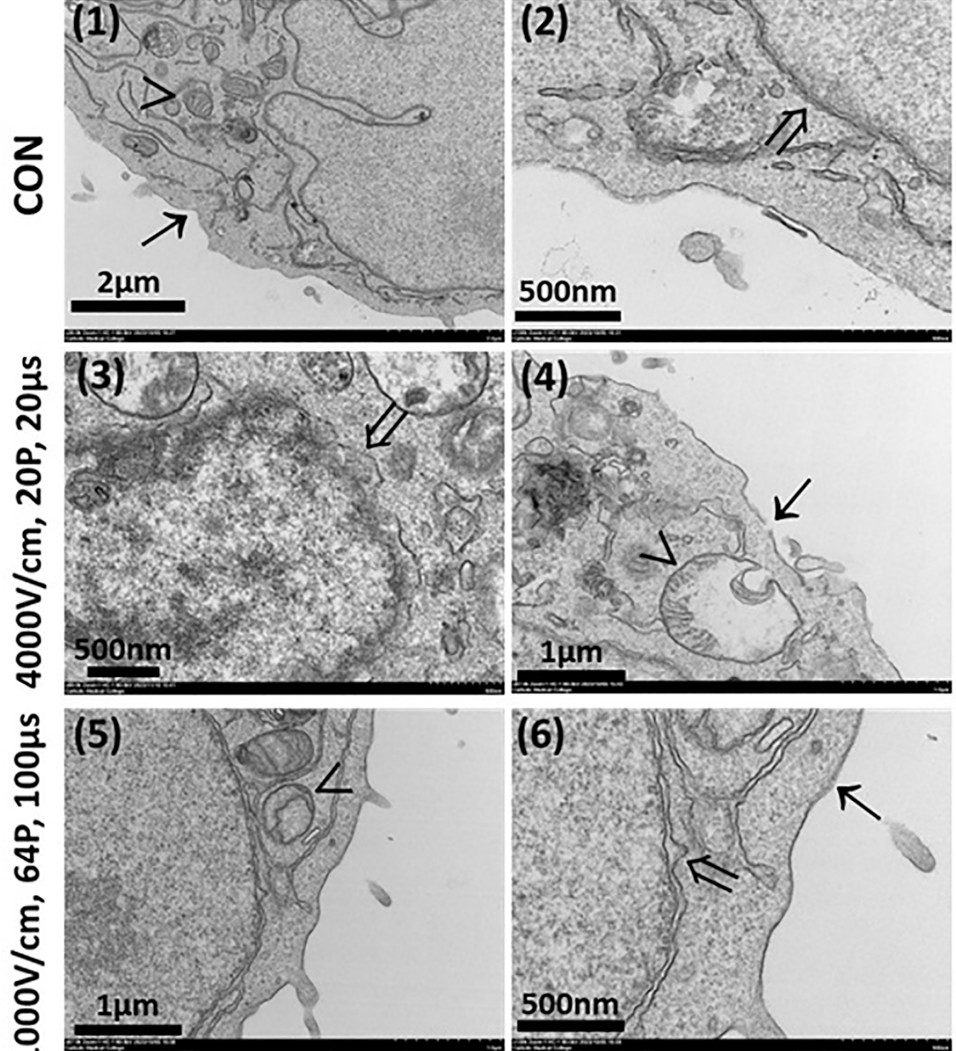

**Fig 7. *in vitro* ultrastructural analysis via transmission electron microscopy.** (1 & 2) Control cells showed intact plasma membranes, nuclear membranes, and mitochondria. (3 & 4) HEF caused severe plasma membrane rupture, substantial nuclear membrane inflation, and extensive mitochondrial damage. (5 & 6) LEF induced moderate plasma membrane disruption, mild nuclear membrane stress, and partial mitochondrial damage. Arrows indicate structures: (→) plasma membrane, (→) nuclear membrane, and (>) mitochondria. Scale bars represent magnification as indicated.

## Discussion

In this study, we demonstrated that high electric field intensity of 4000 V/cm overcomes critical limitations of conventional IRE by achieving complete cellular ablation through extensive damage to both membrane and intracellular compartments. Our iso-energetic comparison revealed that field intensity, not total electrical energy, is the primary determinant of treatment outcome, addressing the fundamental challenge of tissue heterogeneity-induced electric field distribution distortions.

Typical IRE has demonstrated considerable potential as a non-thermal ablative technique for cancer treatment [22]. However, its clinical applications remain limited owing to critical shortcomings Although conventional IRE can induce intracellular organelle injury under certain conditions, accumulating evidence indicates that such effects are often inconsistent and highly dependent on local electric-field distribution. As a result, typical IRE protocols predominantly induce

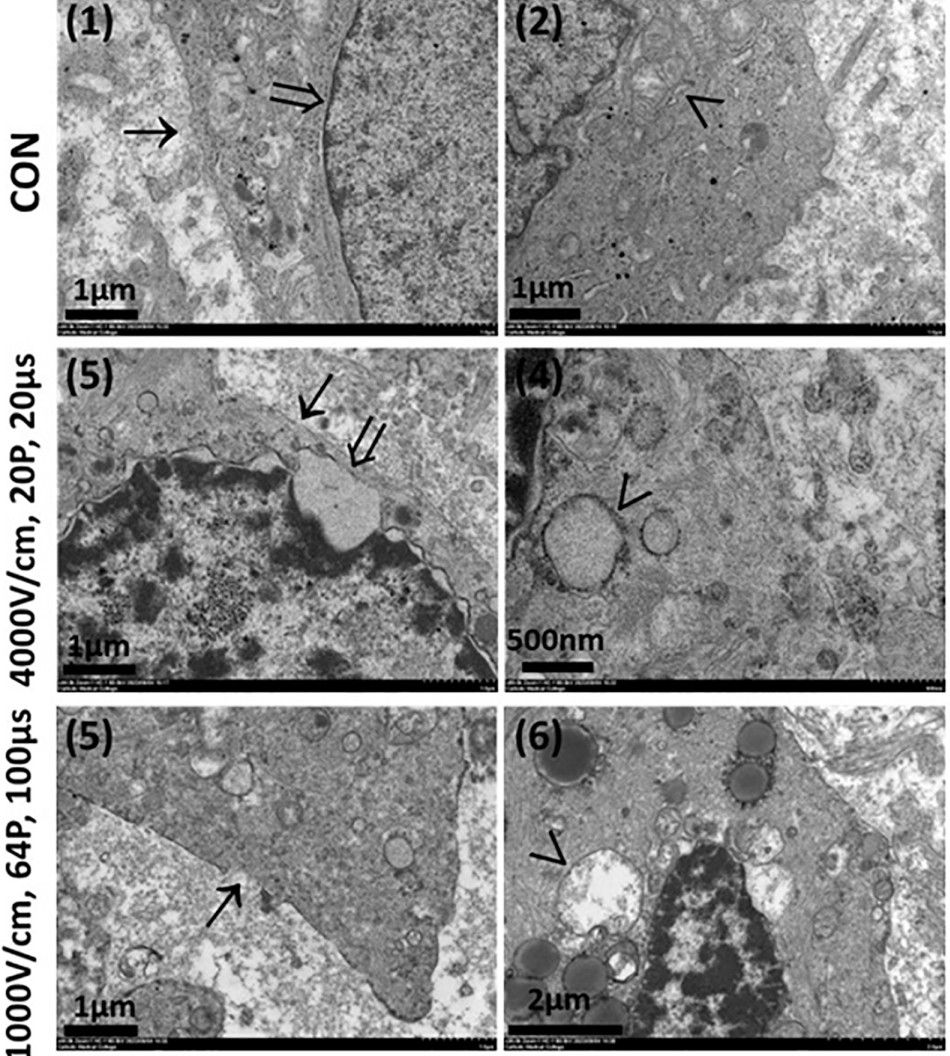

**Fig 8. *in vivo* ultrastructural analysis via transmission electron microscopy.** (1 & 2) Control tissues showed intact plasma membranes, nuclear membranes, and mitochondria. (3 & 4) HEF caused severe plasma membrane rupture, pronounced nuclear membrane inflation, and mitochondrial damage. (5 & 6) LEF caused substantial plasma membrane rupture and mitochondrial damage. Arrows indicate structures: (→) plasma membrane, (→) nuclear membrane, and (>) mitochondria. Scale bars represent magnification as indicated.

plasma-membrane permeabilization while frequently failing to achieve uniform or reliable subcellular damage including mitochondrial dysfunction, nuclear disruption, and cytoskeletal collapse [23]. This incomplete damage increases the likelihood of partial cell death owing to electric field sinks from tissue heterogeneity in structure and conductivity [8,24], as intact nuclei activate repair mechanisms, leading to tumor recurrence or metastasis [25].

Previous studies have shown that heterogeneity in tissue conductivity can distort electric-field distribution [8,26], generating regions where the field strength falls below the irreversible electroporation threshold. Cells located in these sub-threshold regions may survive the procedure, contributing to incomplete ablation. This observation exemplifies a critical limitation of IRE. Surviving cancer cells exhibit increased motility and resistance to subsequent therapies, complicating long-term treatment outcomes [24,27]. Similarly, some studies indicated that incomplete damage from IRE could initiate

tumor recurrence, particularly when the nuclear compartment remains unaffected [8,28]. These findings suggest that current IRE protocols are insufficient for complete tumor eradication and that residual cancer cells alter biological behavior and increase the risks of recurrence and metastasis.

Furthermore, some studies demonstrated that residual cells after IRE exhibit elevated expression of heat HSPs and anti-apoptotic genes, enhancing their ability to survive and adapt to harsh microenvironments [29]. This finding aligns with those of a study indicating that typical IRE does not sufficiently disrupt intracellular compartments, allowing some cells to recover and proliferate [30]. Such residual cells survive and activate EMT pathways, increasing their metastatic potential [32]. These studies indicate that targeting the nuclear membrane and intracellular structures is critical to achieving complete cell destruction and reducing the risks of recurrence and metastasis.

The nuclear membrane is critical in crucial cellular processes, including cell cycle regulation, DNA repair, and apoptosis signaling [20]. Therefore, damaging the nuclear membrane directly inhibits cellular repair and survival mechanisms, ensuring comprehensive cell death. This study explored the application of high electric field-based IRE as a novel approach to overcoming the limitations of typical IRE protocols by targeting the cell membrane, the nuclear membrane, and intracellular structures.

Simulation results confirmed that voltages ranging from 1,600–2,000 V generate electric field strengths exceeding 4,000 V/cm, sufficient for nuclear electroporation while minimizing sparking. Following these simulations, experimental validation confirmed that HEF effectively induced substantially greater damage to the nucleus and mitochondria than LEF. HEF resulted in a threefold increase in DNA damage, demonstrating the enhanced penetration of the electric field gradient into intracellular structures [33]. Additionally, HEF suppressed the expression of DNA repair and apoptotic markers, such as ATR, BRCA1, and p53, suggesting that the damage overwhelmed repair mechanisms, leading to necrosis [33]. In contrast, LEF maintained the expression of these markers, activating stress responses but limiting the extent of intracellular damage [24]. These findings indicate the importance of strong electric fields for the effective targeting of intracellular structures.

Moreover, HEF amplified oxidative stress, a crucial mechanism driving cell death. HEF caused a marked increase MMP disruption and $H_2O_2$ production, demonstrating the induction of oxidative stress in cellular damage [34]. In contrast, LEF induced comparable MMP disruption but substantially less $H_2O_2$ production, indicating that electric field intensity is a critical determinant of ROS-mediated damage. These results support the hypothesis that HEF enhances cancer cell ablation by amplifying oxidative stress pathways.

TEM revealed differences in structural damage induced by LEF and HEF. HEF caused severe disruptions, including plasma membrane rupture, nuclear envelope deformation, and mitochondrial crista loss, consistent with the extensive structural damage required for necrosis [35]. In contrast, LEF induced localized damage while preserving overall cellular structure, suggesting that some cancer cells may survive and increase recurrence risk [24]. HEF addresses these limitations by inducing comprehensive intracellular damage, ensuring complete cancer cell eradication.

In interpreting membrane-level responses, however, it is important to clarify the role of resistance measurements. Because the cell suspension used in these experiments is below the threshold of $1.0 \times 10^8$ cells/mL required to detect membrane-specific impedance changes [36], the resistance decreases observed after pulsing primarily reflect temperature-dependent shifts in bulk medium conductivity rather than intrinsic membrane alterations. The greater resistance change measured under HEF, despite equal total electrical energy delivery, is consistent with the $E^2$-dependent nature of Joule heating and the different temporal profiles of the two waveforms: HEF produces higher instantaneous power and localized heating, whereas LEF distributes energy across longer pulses, allowing greater thermal diffusion. Thus, resistance measurements should be interpreted as indicators of heating dynamics rather than membrane permeability, which is more accurately reflected by CMP and $Ca^{2+}$ influx.

To corroborate these intracellular ultrastructural changes in an *in vivo* test, IREs were performed on lung tumor tissues. As observed *in vitro*, the tissues subjected to HEF exhibited severe plasma membrane rupture, nuclear membrane

deformation, and marked mitochondrial damage, including crista loss. These changes were consistent with necrotic cell death pathways. In contrast, LEF resulted in less severe structural damage, with localized plasma membrane disruptions and partial mitochondrial crista loss; however, the nuclear membrane remained largely intact. These *in vivo* findings align with those of *in vitro*, confirming that HEF induces more extensive and irreversible intracellular damage than LEF, making it more effective for complete tumor eradication.

This study emphasizes the therapeutic potential of HEF in overcoming the limitations of typical membrane-focused IRE. By targeting intracellular structures, including the nucleus, HEF provides a robust alternative capable of mitigating clinical challenges such as recurrence and metastasis. However, this study had some limitations. First, this study did not include a direct comparison with existing high electric field, long-pulse protocols, such as 3,000 V/cm with 100 µs pulses. The absence of such comparisons limits the ability to comprehensively evaluate the relative advantages of short-pulse, high electric field-IRE protocols [37]. Second, the findings were based on *in vitro* experiments and simulations, which may not fully replicate the complexities of *in vivo* systems. Without *in vivo* validation, the broader impacts of HEF on the tumor microenvironment, immune response, and adjacent healthy tissues remain unclear. In addition, the *in vivo* component of this study was intentionally limited to TEM-based ultrastructural analysis. Because the approved animal protocol focused specifically on assessing subcellular structural changes following IRE, additional in vivo evaluations—such as tumor volume monitoring, H&E staining or necrosis quantification, and assessments of systemic responses including inflammation or weight changes—were not included. These parameters are critical for evaluating treatment efficacy and safety, and they will be incorporated into future preclinical studies designed to fully characterize the therapeutic impact of high electric field IRE in vivo. Another limitation of this study is the absence of direct thermal measurements during high-field IRE. Although the short pulse durations used in this protocol (10–40 is) are generally associated with minimal Joule heating and are commonly classified as non-thermal electroporation conditions, localized temperature rises cannot be completely excluded, particularly at high electric field strengths. Thermal imaging, intratumoral thermometry, and histological markers of thermal injury were not included in the approved animal protocol and therefore were not assessed in vivo. These measurements are essential to fully evaluate the clinical safety of high-field IRE, and future studies will incorporate real-time thermal monitoring to distinguish thermal and non-thermal contributions to tissue damage more conclusively. Finally, this study focused on short-term effects, including cellular damage and oxidative stress, and did not address long-term outcomes such as tissue remodeling, immune modulation, or recurrence risks. These limitations underscore the need for additional studies to fully establish the safety, efficacy, and clinical applicability of HEF.

This study demonstrates that HEF effectively targets the nuclear membrane and intracellular structures, overcoming the limitations of typical IRE. This approach enables more comprehensive tumor cell eradication, substantially reducing the likelihood of recurrence and metastasis. The findings establish HEF as a promising foundation for developing next-generation IRE protocols tailored to achieve improved clinical outcomes.

## Acknowledgments

The authors would like to thank C.Y. and M.Y. for their assistance with the experiments.

## Author contributions

**Conceptualization:** Hong Bae Kim, Sung Bo Sim.

**Data curation:** Joon-Mo Yang.

**Formal analysis:** Hong Bae Kim.

**Funding acquisition:** Joon-Mo Yang, Sung Bo Sim.

**Investigation:** Hong Bae Kim.

Methodology: Hong Bae Kim.

Resources: Joon-Mo Yang, Sung Bo Sim.

Software: Jin Young Youm.

Validation: Hong Bae Kim, Sung Bo Sim.

Writing – original draft: Hong Bae Kim.

Writing – review & editing: Hong Bae Kim.

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
