## [Decision Letter · Decision Letter 0]

12 Sep 2025

PONE-D-25-30916

High-intensity irreversible electroporation targeting intracellulafigr structures enhances tumor ablation in lung cancer models

PLOS ONE

Dear Dr. Sim,

Thank you for submitting your manuscript to PLOS ONE. After careful consideration, we feel that it has merit but does not fully meet PLOS ONE’s publication criteria as it currently stands. Therefore, we invite you to submit a revised version of the manuscript that addresses the points raised during the review process.

We look forward to receiving your revised manuscript.

Kind regards,

Roy P. Planalp, Ph.D.

Academic Editor

PLOS ONE

Journal Requirements:

2. To comply with PLOS One submissions requirements, in your Methods section, please provide additional information regarding the experiments involving animals and ensure you have included details on (1) methods of sacrifice, (2) methods of anesthesia and/or analgesia, and (3) efforts to alleviate suffering.

“This research was funded by the National Research Foundation of Korea, grant number 2022R1F1A1075102 and 2022R1F1A1072398, and by Tech Incubator Program for Startup (RS-2023-00303400) of Ministry of SMEs and Startups.”

6. Please amend either the title on the online submission form (via Edit Submission) or the title in the manuscript so that they are identical.

Additional Editor Comments:

As you see, the reviewers are recommending substantial changes, which are somewhat consistent amongst the two.  They both find some untenable assertions being made for which they present literature as support.  Further they call out some of your citations as not relevant.

The following items must be addressed in a resubmission:

1. Any items questioning the vocabulary and terms used in the paper.

2. Directly, any challenge of a conclusion or statement which includes citations of other relevant papers.  Do state clearly whether you agree with cited items that contradict your findings or assertions, and revise accordingly.

3. Suggestions for organization and presentation, e.g. request to add a table from Reviewer 1, or suggestions to explain rationale of the studies.

Suggestions to do additional experiments may be considered with discretion.  If you don't collect any additional data, please state clearly to each suggestion your reason for decline.

Reviewers' comments:

Reviewer's Responses to Questions

**Comments to the Author**

1. Is the manuscript technically sound, and do the data support the conclusions?

Reviewer #1: Yes

Reviewer #2: No

2. Has the statistical analysis been performed appropriately and rigorously? 

Reviewer #1: Yes

Reviewer #2: Yes

3. Have the authors made all data underlying the findings in their manuscript fully available?

The PLOS Data policy requires authors to make all data underlying the findings described in their manuscript fully available without restriction, with rare exception (please refer to the Data Availability Statement in the manuscript PDF file). The data should be provided as part of the manuscript or its supporting information, or deposited to a public repository. For example, in addition to summary statistics, the data points behind means, medians and variance measures should be available. If there are restrictions on publicly sharing data—e.g. participant privacy or use of data from a third party—those must be specified.requires authors to make all data underlying the findings described in their manuscript fully available without restriction, with rare exception (please refer to the Data Availability Statement in the manuscript PDF file). The data should be provided as part of the manuscript or its supporting information, or deposited to a public repository. For example, in addition to summary statistics, the data points behind means, medians and variance measures should be available. If there are restrictions on publicly sharing data—e.g. participant privacy or use of data from a third party—those must be specified.requires authors to make all data underlying the findings described in their manuscript fully available without restriction, with rare exception (please refer to the Data Availability Statement in the manuscript PDF file). The data should be provided as part of the manuscript or its supporting information, or deposited to a public repository. For example, in addition to summary statistics, the data points behind means, medians and variance measures should be available. If there are restrictions on publicly sharing data—e.g. participant privacy or use of data from a third party—those must be specified.requires authors to make all data underlying the findings described in their manuscript fully available without restriction, with rare exception (please refer to the Data Availability Statement in the manuscript PDF file). The data should be provided as part of the manuscript or its supporting information, or deposited to a public repository. For example, in addition to summary statistics, the data points behind means, medians and variance measures should be available. If there are restrictions on publicly sharing data—e.g. participant privacy or use of data from a third party—those must be specified.

Reviewer #1: Yes

Reviewer #2: No

4. Is the manuscript presented in an intelligible fashion and written in standard English?

Reviewer #1: Yes

Reviewer #2: Yes

5. Review Comments to the Author

Reviewer #1: The manuscript entitled “High-intensity irreversible electroporation targeting intracellulafigr structures enhances tumor ablation in lung cancer models” presents an innovative and timely study investigating the effects of high-intensity irreversible electroporation (IRE) on intracellular structures, particularly nuclear membranes and mitochondria, in lung cancer models. The work combines simulation, in vitro experiments, and preliminary in vivo validation, offering significant insights into potential improvements in IRE-based tumor ablation.

Strengths:

The study addresses a critical limitation of conventional IRE, i.e., incomplete intracellular damage leading to possible tumor recurrence.

The combination of computational modeling, in vitro assays, and in vivo ultrastructural analysis provides a comprehensive assessment of high-field effects.

The results are compelling, demonstrating that high electric field intensities induce significant mitochondrial dysfunction, oxidative stress, and nuclear membrane deformation, which could potentially improve tumor eradication.

The manuscript is generally well organized.

Areas for Improvement:

1. Title Typographical Error:

The title contains a typographical error:

“intracellulafigr” → should be “intracellular”

2. Clarification of Energy Equivalence:

The manuscript frequently compares low electric field (LEF) and high electric field (HEF) conditions, stating that they delivered the “same total energy.” However, it is not entirely clear how this equivalence was calculated (e.g., purely Joule heating, energy per pulse, pulse shape effects). I recommend including a table summarizing pulse conditions and explicit calculations of delivered energy.

3. Statistical Details:

Although statistical analyses are reported, several figures lack precise p-values, and it is unclear which post hoc tests were applied after ANOVA. Please clarify statistical methods and ensure consistency across all figures.

4. In Vivo Data Expansion:

The in vivo data are limited to TEM analysis. It would significantly strengthen the manuscript to include: Tumor volume measurements or necrosis quantification post-treatment, Histological analyses beyond TEM (e.g., H&E staining), Data on potential side effects or systemic responses (e.g., inflammation, weight loss)

5. Thermal Considerations:

The discussion briefly mentions temperature increases during high-field IRE. Given that high electric fields might cause localized heating in tissues, additional discussion or in vivo measurements of thermal effects would be valuable for assessing clinical safety.

6. Terminology Consistency:

The manuscript switches between terms like HEF, high-voltage IRE, and high electric field. For clarity, please adopt consistent terminology throughout.

7. Language and Minor Errors:

Besides the title typo, minor wording issues are present,

e.g.: “The viabilities of 4000 and 5000 V/cm were almost similar each other…” → should read: “The viabilities at 4000 and 5000 V/cm were similar to each other.”

In several places, “Data is represented…” should read: “Data are represented…”

Please proofread the manuscript carefully for minor grammatical errors and ensure consistent spelling of technical terms (e.g., spell out “NO” as nitric oxide on first mention).

Questions for the Authors:

1. Could you clarify precisely how “same total energy” was calculated for LEF and HEF conditions?

2. Did you measure tissue temperatures in vivo to confirm that high-field protocols remain non-thermal in tissue?

3. Were any physiological responses (e.g., muscle contractions) observed during in vivo HEF pulsing?

4. Why were in vivo analyses limited to TEM without quantifying macroscopic tumor response?

5. The data suggest suppressed transcriptional activity under HEF (e.g., low p53, ATR). Does this indicate necrosis as the dominant cell death pathway, and were any necrosis-specific markers assessed?

6. Have you considered potential systemic toxicity from high oxidative stress observed in HEF-treated cells?

7. Did you assess potential effects on normal lung tissue adjacent to treated tumors?

8. How does your proposed HEF protocol compare to existing high-frequency IRE (HFIRE) approaches in terms of efficacy and safety?

9. Was current measured during the electroporation procedures, particularly in vivo, to monitor tissue conductivity changes or electrical dose delivered? If so, could you report those results?

Overall, this is an interesting and promising manuscript addressing important challenges in cancer ablation therapy. The study would benefit from clarifying methodological details, expanding in vivo analyses, and addressing minor language issues. I look forward to the authors’ revisions.

Reviewer #2: Summary:

In this submission, Kim et al. describe a method to increase the lethality of IRE by compressing the energy of pulses into shorter pulse widths of higher intensity. It is difficult to draw conclusions from the data as the choice of parameters seems arbitrary, with many comparisons made between 4000 V of 20p at 20 us and 1000 Vof 64p of 100 us, which is not conventional IRE and the normalization was not rationalized. Since so many parameters were varied between two only two groups, the outcomes can be due to voltage, pulse number, or pulse width. Further, there are many incorrect statements with improper citations that do not support the statements, some of which talk about IRE with papers that are not electroporation at all.

Comments:

1. There is a typo in the submission title. Not sure if this will cause a problem.

2. “Spark-free” is mentioned a few times. Sparking was not necessarily a problem with IRE and H-FIRE previously, and the terminology makes it seem like you are overcoming this problem which isn’t prevalent.

3. The introduction heavily cites review articles, which do not entirely support the statements being made, particularly with statements that IRE is membrane-focused and minimally affects intracellular structures. Higher frequency pulses may not be as attenuated by the cell membrane impedance, but there is a trade off with pore formation decreasing the membrane resistance and increasing the capacitance. If the cell membrane experiences a pulse, it does not know 2 µs into the pulse if the pulse will end or if it will continue for another 98 µs. For IRE and H-FIRE during those initial 2 µs, the electric field distribution within the cell and outside should theoretically be the same. Ignoring that IRE can physically hemorrhage the cell which would affect all the contents, IRE will also damage intracellular structures, but perhaps the ratio of membrane disruption to intracellular disruption is different between different pulse widths. For the same electric field strength (i.e., 2000 V/cm), IRE will affect intracellular structures more than H-FIRE. However, in the context of this paper, the pulse widths are not short enough to be beyond the beta dispersion caused by membrane capacitance.

4. “Studies have shown that residual cancer cells, after membrane-focused IRE, exhibit increased motility, elevated expression of heat shock proteins (HSPs), and enhanced resistance to subsequent therapies[12].” Citation 12 is not an electroporation paper and does not support this claim. There is literature demonstrating the opposite, that electroporation decreases genetic expression associated with metastasis (doi: 10.1016/j.bioelechem.2025.109036.) and decreases motility of cells following treatment in brain cancers (doi: 10.1016/j.bioelechem.2025.109036. , 10.1016/j.bioelechem.2025.109005).

5. Citations 15 and 16 for high-electric-field IRE are nanosecond and H-FIRE, not IRE.

6. H-FRE has only been briefly used in human tumors, but there are in vivo studies using H-FIRE >2,300 V/cm (https://doi.org/10.34133/bmef.0169, 10.1109/TBME.2024.3468159, 10.1016/j.ebiom.2019.05.036).

7. Was the conductivity of the media measured, because my understanding is that cell culture media (i.e., RPMI, DMEM) is roughly ~1 S/m at room temperature and 1.2 S/m at 37C (10.1115/1.4053595). Also, how would half filling the cuvette reduce the sparking risk? My understanding is that sparking is due to electric field or current density intensity, which wouldn’t really change if it was ½ as tall? The electric field would be roughly the same, but the volume would be half, meaning the current should be roughly half. However, the current scale linearly with the height, meaning the current density should be the same no matter the height? If it was experimentally observed, then that is interesting and possibly due to heating?

8. Be careful using the word significant when statistics were not done. There were not statistical analyses. E.g., “At higher voltages of 1600 V and 2000 V, significant intensification of the electric field was evident, reaching critical strengths exceeding 4000 V/cm, indicated by the red color zones”

9. “Viabilities of 20 and 40 pulses were similar due to reaching saturation effect.” This is contradictory of previous IRE and H-FIRE results that have seen a much higher saturation (10.1016/j.bioelechem.2023.108580).

10. There are many parameters being adjusted without much rationale for choices. The results needs to clearly explain why these parameters are being adjusted and general conclusion for that results section. It just seems like a lot of different things were done without a coherent story, reasoning for why, and what the results mean.

11. For the resistance changes, what is the control resistance. The figure should clearly indicate if it is an increase or decrease in resistance. Also, the cell suspensions presented are too low of cell density to see changes in resistance due to electroporation, so you are most likely measuring conductivity changes due to temperature, so “indicates that HEF may induce strong membrane polarization and destabilization, potentially increasing susceptibility to additional stress” may not be accurate. Even in super dense cell suspensions, only small changes in membrane resistance have been measured, but 1x10^5 cells /ml is orders of magnitude much too low than the >5x10^7 cell/ml to see these effects (https://doi.org/10.1007/978-3-319-26779-1_164-1,
https://doi.org/10.1529/biophysj.104.048975).

12. For gene expression analysis, did the house keeper gene significantly change with treatment. If so, then you know that translation was impaired due to cell death. Using a gene that mediates cell death is not appropriate, as it is involved in the processes being induced.

13. “Typical IRE protocols primarily focus on permeabilizing the cell membrane, leaving the nuclear membrane largely intact [8]” Though the treatment done is not exactly IRE (100 µs for 100 pulses or NanoKnife 90 µs for 90 pulses), the results presented demonstrate that IRE does induce significant nuclear damage, at a lower electric field than used clinically (10.2478/raon-2025-0011).

14. This statement is incorrect and not properly cited. Citation 24 is not cancer and does not mention nucleus in the paper: “A previous study reported that up to 30% of cancer cells treated with membrane-focused IRE survive because the nuclear membrane protects genetic material and crucial cellular functions, enabling recovery and survival [24]”. They demonstrate that heterogeneous tissue conductivity causes distortions of the electric field that can drop the lethal electric field below the threshold for areas within the tissue.

6. PLOS authors have the option to publish the peer review history of their article (what does this mean?). If published, this will include your full peer review and any attached files.). If published, this will include your full peer review and any attached files.). If published, this will include your full peer review and any attached files.). If published, this will include your full peer review and any attached files.

...

Reviewer #1: No

Reviewer #2: No

---

## [Author Response · Author response to Decision Letter 1]

8 Dec 2025

Reviewer 1

1. Title Typographical Error:

The title contains a typographical error:

“intracellulafigr” → should be “intracellular”

Author Response:

We revised the title correctly.

Revision:

High-intensity irreversible electroporation targeting intracellular structures enhances tumor ablation in lung cancer models

2. Clarification of Energy Equivalence:

The manuscript frequently compares low electric field (LEF) and high electric field (HEF) conditions, stating that they delivered the “same total energy.” However, it is not entirely clear how this equivalence was calculated (e.g., purely Joule heating, energy per pulse, pulse shape effects). I recommend including a table summarizing pulse conditions and explicit calculations of delivered energy.

Author Response:

We appreciate your insightful comment. To clarify how energy equivalence between the LEF and HEF conditions was established, we have now added a detailed description and a new table summarizing the pulse parameters and corresponding energy indices.

Specifically, the “energy index” (V²·t·N) was used to represent the relative total electrical energy delivered per treatment, where V is the electric field strength (V/cm), t is the pulse width (s), and N is the number of pulses. This index reflects the proportional relationship to Joule energy (E ∝ V²·t·N/R), assuming a constant impedance across conditions.

The new table (Table 1) presents the calculated energy indices for all pulse settings applied at 1 Hz under both HEF and LEF protocols. This addition improves transparency and allows direct comparison of the total energy delivered under different electric field conditions.

Voltage (V/cm) Pulse width (µs) Energy index V2·t·N (V2·sec/cm2)

Pulse number

10 20 40

HEF 1000 10 100 200 400

2000 10 400 800 1600

3000 10 900 1800 3600

4000 10 1600 3200 6400

20 0 6400 0

40 0 12800 0

5000 10 2500 5000 10000

6000 10 3600 0 0

LEF Pulse number

64

1000 100 6400

Applied all to be frequency 1Hz

3. Statistical Details:

Although statistical analyses are reported, several figures lack precise p-values, and it is unclear which post hoc tests were applied after ANOVA. Please clarify statistical methods and ensure consistency across all figures.

Author Response:

Thank you for this valuable comment. In response, we thoroughly revised our statistical reporting for improved clarity and consistency. Specifically:

All figures (Fig. 2–Fig. 6) were updated so that ANOVA-based comparisons are clearly indicated using the symbols #, ##, and ###, while t-test–based comparisons are marked using *, **, and ***.

All figure legends were rewritten to explicitly state the statistical method used (ANOVA with Tukey’s HSD post hoc test or unpaired t-test), along with the meaning of each symbol and the sample size (n).

The Statistical Analysis section in the Materials and Methods has been revised to clearly describe:

the use of one-way ANOVA followed by Tukey’s HSD post hoc test for multiple-group comparisons,

the use of unpaired two-tailed t-tests for two-group comparisons, and

the symbol system applied in the updated figures.

Exact p-values, or t statistics, and degrees of freedom are now included in Supplementary Table S1 for full transparency.

Revision:

Figure:

Figure 2.

Figure 3.

Figure 4.

Figure 5.

Figure 6.

Legends:

Fig 2. Cell Viability and apoptosis under various IREs in A549 cells. (1) Cell viability after various electric field intensity treatment with pulse counts of 10, 20, and 40 pulses fixed at 20μs. (2) Cell viability at a fixed electric field of 4000 V/cm and 20 pulses, with pulse durations of 10, 20, and 40 μs. (3) Comparison of cell viability between 1000 V/cm and 4000 V/cm. (4) Apoptosis across increasing electric field strengths at a fixed pulse duration of 10 μs and pulse number of 20. (5) Apoptosis at electric field strength of 4,000 V/cm and 20 pulses with varying pulse durations of 10, 20, and 40 μs. (6) Comparison of apoptosis between 1000 V/cm and 4000 V/cm. Data are represented as mean ± SEM (n =3). Multiple-group comparisons were analyzed using one-way ANOVA followed by Tukey’s HSD post hoc test, with significance indicated as: ###p < 0.001. Two-group tests were analyzed using unpaired two-tailed t-tests, with significance indicated as: *p < 0.05, **p < 0.01, ***p < 0.001.

Fig 3. The electrophysiological responses of A549 cells. (1) Change in electrical resistance (Ω) in response to different electric field strengths. (2) Cytomembrane potential (CMP). (3) Cytosolic calcium concentration. Data is represented as mean ± SEM (n = 3). Multiple-group comparisons were analyzed using one-way ANOVA followed by Tukey’s HSD post hoc test, with statistical significance indicated as: ###p < 0.001; N.S, not significant. Two-group tests were analyzed using unpaired two-tailed t-tests, with significance indicated as: *p < 0.05, **p < 0.01, ***p < 0.001.

Fig 4. Impact of variable IRE parameters on mitochondrial dysfunction and oxidative stress. (1) Mitochondrial membrane potential (MMP). (2) H2O2 production. (3) NO production. Data is represented as mean ± SEM (n = 3). Multiple-group comparisons were analyzed using one-way ANOVA followed by Tukey’s HSD post hoc test, with statistical significance indicated as: ###p < 0.001; N.S, not significant. Two-group tests were analyzed using unpaired two-tailed t-tests, with significance indicated as: *p < 0.05, **p < 0.01, ***p < 0.001.

Fig 5. Evaluation of total DNA damage in A549 cells. Total DNA damage was quantified by the combined percentage of cells positive for phosphorylated ATM, and H2A.X. Data represent mean ± SEM (n = 3). Multiple-group comparisons were analyzed using one-way ANOVA followed by Tukey’s HSD post hoc test, with statistical significance indicated as: ###p < 0.001. Two-group tests were analyzed using unpaired two-tailed t-tests with significance indicated as: **p < 0.01, ***p < 0.001.

Fig 6. Differential gene expression analysis in A549 cells. Real-time PCR showing the relative mRNA expression levels normalized to the reference gene ClAO1, involved in DNA damage sensing (ATM (1) and ATR (2)), stress response coordination (CHK1 (3), CHK2 (4)), DNA repair (PARP1 (5), and BRCA1 (6), and apoptosis (p53 (7) and BAX (8)). Data are presented as mean ± SEM (n = 3). Multiple-group comparisons were analyzed using one-way ANOVA followed by Tukey’s HSD post hoc test, with statistical significance indicated as: #p < 0.05, ##p < 0.01, ###p < 0.001; N.S, not significant. Two-group tests were analyzed using unpaired two-tailed t-tests, with significance indicated as: *p < 0.05, **p < 0.01, ***p < 0.001.

Statistical analysis

All experiments were repeated at least thrice, and data were expressed as the mean ± standard deviation. For comparisons involving more than two groups, statistical significance was assessed using a one-way analysis of variance (ANOVA) followed by Tukey’s HSD post-hoc test. For comparisons involving only two groups, an unpaired two-tailed Student's t-test was used. All statistical analyses were performed using Excel software. To clearly distinguish between test types in the figures, ANOVA/Tukey post hoc results are indicated by symbols (#, ##, ###), whereas **t-test results are indicated by asterisk (*, **, ***). Statistical significance thresholds were defined as: #p < 0.05, ##p < 0.01, ###p < 0.001 (ANOVA + Tukey HSD); *p < 0.05, **p < 0.01, and ***p < 0.001(t-test).

4. In Vivo Data Expansion:

The in vivo data are limited to TEM analysis. It would significantly strengthen the manuscript to include: Tumor volume measurements or necrosis quantification post-treatment, Histological analyses beyond TEM (e.g., H&E staining), Data on potential side effects or systemic responses (e.g., inflammation, weight loss)

Author Response:

Thank you for this insightful suggestion. We agree that additional in vivo assessments, including tumor volume tracking, histological analyses, and evaluation of systemic responses, would further strengthen the manuscript. However, the in vivo component of this study was conducted with a specific and limited objective: to evaluate ultrastructural changes in tumor tissue following IRE, using TEM as the primary analytical method.

The approved IACUC protocol (CMC 20-004) covered only a minimal exploratory design involving

subcutaneous tumor formation,

delivery of two IRE electric field conditions (4,000 V/cm and 1,000 V/cm), and

subsequent TEM-based ultrastructural evaluation.

Comprehensive tumor-volume studies, extended histology, or systemic toxicity analyses were not included in the approved protocol, and therefore cannot be retrospectively added.

To address your comment, we have added a clear statement in the Discussion section acknowledging this limitation and outlining our plan to incorporate full preclinical assessments—such as tumor regression analysis, H&E staining, quantification of necrosis, and systemic response monitoring—in a follow-up investigation.

Revision:

Discussion:

In addition, the in vivo component of this study was intentionally limited to TEM-based ultrastructural analysis. Because the approved animal protocol focused specifically on assessing subcellular structural changes following IRE, additional in vivo evaluations—such as tumor volume monitoring, H&E staining or necrosis quantification, and assessments of systemic responses including inflammation or weight changes—were not included. These parameters are critical for evaluating treatment efficacy and safety, and they will be incorporated into future preclinical studies designed to fully characterize the therapeutic impact of high electric field IRE in vivo.

5. Thermal Considerations:

The discussion briefly mentions temperature increases during high-field IRE. Given that high electric fields might cause localized heating in tissues, additional discussion or in vivo measurements of thermal effects would be valuable for assessing clinical safety.

Author Response:

Thank you for this important comment. We agree that thermal effects are a critical aspect of evaluating the safety of high-field IRE. In this study, our primary objective was to investigate the mechanistic, non-thermal intracellular effects of short-pulse high electric fields, and therefore direct thermal measurements were not included in the in vivo protocol. To address your concern, we have added additional discussion clarifying the thermal characteristics of our short-pulse HEF protocol, referencing prior literature demonstrating minimal Joule heating under similar pulse widths (10–40 μs). We also explicitly acknowledge that the absence of in vivo temperature measurements is a limitation of the current study and outline that future work will incorporate real-time intratumoral thermometry, infrared thermal mapping, and histological indicators of thermal injury to comprehensively assess safety.

Revision:

Discussion:

Another limitation of this study is the absence of direct thermal measurements during high-field IRE. Although the short pulse durations used in this protocol (10–40 μs) are generally associated with minimal Joule heating and are commonly classified as non-thermal electroporation conditions, localized temperature rises cannot be completely excluded, particularly at high electric field strengths. Thermal imaging, intratumoral thermometry, and histological markers of thermal injury were not included in the approved animal protocol and therefore were not assessed in vivo. These measurements are essential to fully evaluate the clinical safety of high-field IRE, and future studies will incorporate real-time thermal monitoring to distinguish thermal and non-thermal contributions to tissue damage more conclusively.

6. Terminology Consistency:

The manuscript switches between terms like HEF, high-voltage IRE, and high electric field. For clarity, please adopt consistent terminology throughout.

Author Response:

Thank you for pointing this out. We agree that consistent terminology is essential for clarity. To address this, we have standardized terminology throughout the manuscript.

Specifically, we now use:

HEF-IRE (High Electric Field Irreversible Electroporation) for high-field conditions, and

LEF-IRE (Low Electric Field Irreversible Electroporation) for comparison conditions.

All instances of “high-voltage IRE,” “high electric field,” “strong electric field,” and other similar variations have been unified under HEF-IRE, and all related abbreviations have been updated in the text, figures, and legends accordingly.

7. Language and Minor Errors:

Besides the title typo, minor wording issues are present,

e.g.: “The viabilities of 4000 and 5000 V/cm were almost similar each other…” → should read: “The viabilities at 4000 and 5000 V/cm were similar to each other.”

In several places, “Data is represented…” should read: “Data are represented…”

Please proofread the manuscript carefully for minor grammatical errors and ensure consistent spelling of technical terms (e.g., spell out “NO” as nitric oxide on first mention).

Author Response:

As you pointed out, we all revised in relation to your concerns.

Questions for the Authors:

(1) Could you clarify precisely how “same total energy” was calculated for LEF and HEF conditions?

Author Response:

Thank you for this important comment. In this study, “same total energy” refers to the electrical energy delivered by a given pulse protocol, which we approximated from the pulse parameters.

Because the electrode geometry, gap, and medium were identical within each experiment, we assumed that the load impedance remained constant. Under this assumption, the Joule energy (W) delivered by a pulse train is proportional to:

W∝V^2×τ×N

where V is the applied voltage, τ is the pulse duration, and N is the number of pulses. For each HEF condition, we selected a corresponding LEF condition such that the product V^2⋅τ⋅Nwas comparable, thereby matching the total electrical energy index between LEF and HEF while varying the electric field strength and pulse width.

We added this calculation as in the “2. Clarification of Energy Equivalence” and in table 1.

(2) Did you measure tissue temperatures in vivo to confirm that high-field protocols remain non-thermal in tissue?

Author Response:

Thank you for this important question. We did not directly measure tissue temperatures in vivo in the present study. The in vivo experiment was designed as an exploratory study with a primary focus on ultrastructural changes observed by TEM, and the approved animal protocol did not include intratumoral thermometry or thermal imaging. We attempted to minimize thermal effects by using short pulse durations of 10–40 μs and a limited number of pulses, which are generally considered within the non-thermal electroporation regime. No macroscopic signs of thermal injury were observed in the treatment tissues during or immediately after IRE application. However, we fully agree that the absence of direct temperature measurements is a limitation.

(3) Were any physiological responses (e.g., muscle contractions) observed during in vivo HEF pulsing?

Author Response:

Thank you for this important question. Because the HEF-IRE pulses used in this study were monophasic, transient muscle contractions were indeed observed during in vivo pulsing. However, these contractions were substantially weaker than those typically associated with longer pulse protocols, owing to the very short pulse durations of 10–40 μs used in our HEF-IRE settings. We did not quantitatively measure muscle activation, as the in vivo experiment was designed primarily for ultrastructural analysis using TEM. Nonetheless, the visible muscle responses were brief and mild, without causing observable motion artifacts or requiring additional stabilization.

(4) Why were in vivo analyses limited to TEM without quantifying macroscopic tumor response?

Aut

---

## [Editor Report · Decision Letter 1]

22 Dec 2025

PONE-D-25-30916R1High-intensity irreversible electroporation targeting intracellular structures enhance tumor ablation in lung cancer modelsPLOS One

Dear Dr. Sim,

Thank you for submitting your manuscript to PLOS ONE. After careful consideration, we feel that it has merit but does not fully meet PLOS ONE’s publication criteria as it currently stands. Therefore, we invite you to submit a revised version of the manuscript that addresses the points raised during the review process.

**I am returning your manuscript for revision without review because I note that points 11-14 of reviewer 2's critique have not been responded in your summary of changes to the manuscript.  One should be aware that reviewers put considerable effort into evaluation, without direct compensation.  To protect reviewer time and keeping with editor support of the reviewers, this issue is to be addressed before considering any further review of changes.  We look forward to a full revision.** ============================================================

We look forward to receiving your revised manuscript.

Kind regards,

Roy P. Planalp, Ph.D.

Academic Editor

PLOS One
---

## [Author Response · Author response to Decision Letter 2]

12 Jan 2026

"We have provided a detailed, point-by-point response to all comments from the Editor and Reviewers (including the responses to Reviewer 2, Points 11–14) in the separate uploaded file labeled 'Response to Reviewers'. Please refer to that document for our full response."

---

## [Decision Letter · Decision Letter 2]

20 Mar 2026

High-intensity irreversible electroporation targeting intracellular structures enhance tumor ablation in lung cancer models

PONE-D-25-30916R2

Dear Dr. Sim,

We’re pleased to inform you that your manuscript has been judged scientifically suitable for publication and will be formally accepted for publication once it meets all outstanding technical requirements.

Kind regards,

Roy P. Planalp, Ph.D.

Academic Editor

PLOS One

Additional Editor Comments (optional):

There's a suggestion to add a reference, from reviewer 1, which you would want to address.  Congratulations on successful work!

Reviewers' comments:

Reviewer's Responses to Questions

**Comments to the Author**

1. If the authors have adequately addressed your comments raised in a previous round of review and you feel that this manuscript is now acceptable for publication, you may indicate that here to bypass the “Comments to the Author” section, enter your conflict of interest statement in the “Confidential to Editor” section, and submit your "Accept" recommendation.

Reviewer #1: All comments have been addressed

Reviewer #2: All comments have been addressed

2. Is the manuscript technically sound, and do the data support the conclusions?

Reviewer #1: Yes

Reviewer #2: Yes

3. Has the statistical analysis been performed appropriately and rigorously? 

Reviewer #1: Yes

Reviewer #2: Yes

4. Have the authors made all data underlying the findings in their manuscript fully available?

The PLOS Data policy requires authors to make all data underlying the findings described in their manuscript fully available without restriction, with rare exception (please refer to the Data Availability Statement in the manuscript PDF file). The data should be provided as part of the manuscript or its supporting information, or deposited to a public repository. For example, in addition to summary statistics, the data points behind means, medians and variance measures should be available. If there are restrictions on publicly sharing data—e.g. participant privacy or use of data from a third party—those must be specified.requires authors to make all data underlying the findings described in their manuscript fully available without restriction, with rare exception (please refer to the Data Availability Statement in the manuscript PDF file). The data should be provided as part of the manuscript or its supporting information, or deposited to a public repository. For example, in addition to summary statistics, the data points behind means, medians and variance measures should be available. If there are restrictions on publicly sharing data—e.g. participant privacy or use of data from a third party—those must be specified.requires authors to make all data underlying the findings described in their manuscript fully available without restriction, with rare exception (please refer to the Data Availability Statement in the manuscript PDF file). The data should be provided as part of the manuscript or its supporting information, or deposited to a public repository. For example, in addition to summary statistics, the data points behind means, medians and variance measures should be available. If there are restrictions on publicly sharing data—e.g. participant privacy or use of data from a third party—those must be specified.requires authors to make all data underlying the findings described in their manuscript fully available without restriction, with rare exception (please refer to the Data Availability Statement in the manuscript PDF file). The data should be provided as part of the manuscript or its supporting information, or deposited to a public repository. For example, in addition to summary statistics, the data points behind means, medians and variance measures should be available. If there are restrictions on publicly sharing data—e.g. participant privacy or use of data from a third party—those must be specified.

Reviewer #1: Yes

Reviewer #2: Yes

5. Is the manuscript presented in an intelligible fashion and written in standard English?

Reviewer #1: Yes

Reviewer #2: Yes

6. Review Comments to the Author

Reviewer #1: The authors have addressed the reviewer comments satisfactorily, and the manuscript has improved significantly. I have no major remaining concerns.

I only suggest a brief clarification in the Methods section regarding the assumption of constant impedance when using the V2.t.N energy index, since tissue conductivity may change during electroporation. A short discussion of this limitation would further strengthen the methodological description.

With this minor clarification, I consider the manuscript suitable for publication.

Reviewer #2: The authors have done a great job answering the questions thoroughly and making the appropriate fixes within the manuscript.

7. PLOS authors have the option to publish the peer review history of their article (what does this mean?). If published, this will include your full peer review and any attached files.). If published, this will include your full peer review and any attached files.). If published, this will include your full peer review and any attached files.). If published, this will include your full peer review and any attached files.

...

Reviewer #1: **Yes:** POOMPAVAI SADASIVAMPOOMPAVAI SADASIVAMPOOMPAVAI SADASIVAMPOOMPAVAI SADASIVAM

Reviewer #2: No

---

## [Editor Report · Acceptance letter]

PONE-D-25-30916R2

PLOS One

Dear Dr. Sim,

I'm pleased to inform you that your manuscript has been deemed suitable for publication in PLOS One. Congratulations! Your manuscript is now being handed over to our production team.

Kind regards,

on behalf of

Dr. Roy P. Planalp

Academic Editor

PLOS One